# Medical Physics Adaptive Radiotherapy (MPART) Fellowship: Bridging the Training Gap in Online Adaptive Radiotherapy

**DOI:** 10.3390/healthcare13243315

**Published:** 2025-12-18

**Authors:** Bin Cai, David Parsons, Mu-Han Lin, Dan Nguyen, Andrew R. Godley, Arnold Pompos, Kajal Desai, Shahed Badiyan, David Sher, Robert Timmerman, Steve Jiang

**Affiliations:** Department of Radiation Oncology, University of Texas Southwestern Medical Center, Dallas, TX 75390, USA

**Keywords:** adaptive radiotherapy, medical physics fellowship, medical physics education

## Abstract

Online adaptive radiotherapy (ART) is rapidly transforming clinical radiation oncology by enabling adaptation of treatment plans based on patient-specific anatomical and biological changes. However, most medical physics training programs lack structured education in ART. To address this critical gap, the Medical Physics Adaptive Radiotherapy (MPART) Fellowship was established at our center to train post-residency or practicing physicists in advanced adaptive technologies and workflows. The MPART Fellowship is a two-year program that provides immersive, platform-specific training in CBCT-guided (Varian Ethos), MR-guided (Elekta Unity), and PET-guided (RefleXion X1) radiotherapy. Fellows undergo modular clinical rotations, hands-on training, and dedicated research projects. The curriculum incorporates competencies in imaging, contouring, online planning, quality assurance, and team-based decision-making. Evaluation is based on the Accreditation Council for Graduate Medical Education competency domains and includes milestone tracking, mentor reviews, and structured presentations. The fellowship attracted applicants from both domestic and international institutions, reflecting strong demand for formal ART training. Out of 22 applications, two fellows have been successfully recruited into the program since 2024. Fellows actively participate in all phases of adaptive workflows and are expected to function at near-attending levels by the second year of their training. Each fellow also leads at least one translational or operational research project aimed at improving ART delivery. Fellows contribute to clinical coverage and lead developmental projects, resulting in presentations and publications at the national and international levels. The MPART Fellowship addresses a vital educational need by equipping medical physicists with the advanced competencies necessary for implementing and leading ART. This program offers a replicable framework for other institutions seeking to advance precision radiation therapy through structured post-residency training in adaptive radiotherapy. As this fellowship program is still in its early phase of establishment, the primary goal of this paper is to introduce the structure, framework, and implementation model of the program. Comprehensive outcome analyses—such as quantitative assessments, fellow feedback, and longitudinal competency evaluations—will be incorporated in future work as additional cohorts complete training.

## 1. Introduction

The online adaptive radiotherapy (ART) has emerged as a transformative paradigm in modern radiation oncology [1,2,3]. With the clinical deployment of advanced image-guided platforms—such as CT or CBCT-based systems (e.g., Varian Ethos, Elekta Evo, United Imaging uRT-linac) and MR-guided systems (e.g., ViewRay MRIdian and Elekta Unity)—clinicians can now tailor radiation treatment right before a fraction starts. These capabilities have demonstrated potential to improve therapeutic ratios, reduce planning target volume margins, and support more personalized radiation delivery strategies [2,3,4,5,6].

The adoption of these technologies is accelerating. Institutions across the world have initiated ART programs, and early clinical outcomes suggest that adaptive workflows may lead to improved local control and reduced toxicity, particularly in disease sites characterized by significant interfractional variability or high sensitivity to normal tissue sparing [4,7,8,9,10,11,12,13,14,15]. More broadly, AI has demonstrated growing potential to enhance the efficiency and accuracy of medical imaging, contouring, and treatment planning. In contemporary ART platforms, AI-enabled tools are increasingly used to support time-sensitive online delineation and re-planning, underscoring the need for structured training that includes AI-aware quality oversight and failure-mode recognition [16]. However, while the technological infrastructure has advanced rapidly, the training of medical physicists has lagged behind this clinical evolution.

Most medical physics residency programs accredited by the Commission on Accreditation of Medical Physics Education Programs (CAMPEP) do not include formal, longitudinal training in online ART. Several barriers contribute to this gap: many training institutions do not yet have access to ART-enabled treatment systems; faculty bandwidth and infrastructure to support specialized ART mentorship are often limited; and residency curricula are already dense, leaving little room to accommodate the hands-on and system-specific learning required by ART. Furthermore, the rapid and continuous evolution of ART technologies complicates the development of a standardized, didactic curriculum. Instead, ART demands experiential learning in high-volume, clinically active environments where physicists can gain practical exposure to real-world planning, adaptation, and delivery challenges.

This educational gap is significant and warrants urgent attention. Online ART workflows demand specialized competencies that are not traditionally included in standard medical physics residency training. These include adaptive contouring, deformable image registration, robust adaptive planning, real-time optimization under time constraints, and immediate quality assurance decision-making. The absence of structured, comprehensive ART training not only poses risks to the safe and effective clinical implementation of adaptive techniques but also impedes their broader adoption and standardization across institutions.

Limited access to advanced ART technologies and platforms further exacerbates the issue, contributing to a shortage of qualified physicists capable of serving as primary quality instructors in adaptive treatment settings. Currently, most ART-specific training is delivered either through vendor-led sessions or informal, on-the-job experiences at institutions that already have ART capabilities. These training opportunities are typically narrow in scope—confined to a specific system, module, or institutional workflow—and lack generalizability. As a result, the dissemination of ART knowledge remains fragmented and inconsistent.

To address this need, we established the Medical Physics Adaptive Radiotherapy (MPART) Fellowship Program. The primary focus of this report is to describe the establishment of a dedicated MPART Fellowship. The structure and experiences outlined here may also provide a useful framework for medical physics residency or graduate training programs seeking to incorporate adaptive radiotherapy (ART) components into their curricula.

Recent ART training opportunities include vendor-provided initial and refresher programs that are typically platform-specific and delivered at vendor headquarters or on-site at client institutions. In addition, some centers offer short courses and workshops, often limited to several days or one week, and select residency programs may provide internal ART elective rotations that are not broadly accessible. These resources are valuable for disseminating knowledge and building familiarity with specific systems. The MPART Fellowship is designed to complement these models by providing a formal, multi-year, multi-modality training pathway with structured competency assessments across CBCT-, MR-, and PET-guided platforms. Unlike short, platform-specific courses, MPART integrates longitudinal, immersive clinical participation in planning, online adaptive coverage, and patient-specific QA, alongside opportunities for translational research. The program is intended to develop future ART leaders and to build upon—not replace—standard CAMPEP residency training. Fellows are also encouraged to disseminate their experience through abstracts and peer-reviewed publications.

## 2. Materials and Methods

### 2.1. Program Goals

The overarching goal of the MPART Fellowship is to develop the next generation of medical physicists who are equipped to lead the clinical implementation, optimization, and innovation of personalized online ART.

The fellowship is designed to deliver structured, in-depth training across two primary domains:Physics knowledge, encompassing technology- and platform-specific expertise in imaging, planning, and delivery of ART;Clinical knowledge, focused on disease site–specific protocols, adaptive decision-making, and multidisciplinary integration in the context of routine patient care.

Objectives within each category are listed in Table 1.

In addition to these core competencies, the program also aims to support the professional development of fellows by fostering skills in communication, teamwork, ethics, and career planning—essential attributes for future leaders in the evolving landscape of radiation oncology.

### 2.2. Curriculum Topics and Hands-On Clinical Integration

In developing the MPART Fellowship, we drew upon a range of authoritative resources related to medical physics education [17,18,19,20,21], ART program implementation, institutional standard operating procedures (SOPs), and prior clinical experience. Several references were particularly instrumental in shaping the program structure and identifying key training focus areas. These include AAPM Report No. 249 [21], which provides guidance for CAMPEP-accredited residency programs; the CAMPEP Residency Standards [19]; the ASTRO white paper on quality and safety considerations for ART [22]; and technical recommendations and strategy documents from NRG Oncology on ART planning and delivery [23]. These resources collectively informed both the curriculum and operational design of the fellowship.

The core training topics covered in the MPART Fellowship includes: (1) ART imaging protocols, encompassing simulation procedures and daily image acquisition techniques; (2) Target delineation and review in the online setting, with emphasis on adaptive contouring principles and anatomical uncertainties; (3) ART planning strategies, spanning both offline initial planning and online reoptimization approaches, including the use of AI-assisted tools, adaptive dose objectives, and replanning criteria; (4) Workflow checklists and structured decision-making frameworks for real-time plan approval during adaptive sessions; (5) Adaptive quality assurance (QA) procedures, such as secondary dose verification and post-treatment review; (6) Dose accumulation and evaluation across treatment fractions; (7) Participation in incident learning systems, focusing on ART-specific safety concerns and mitigation strategies; and (8) Process-based risk analysis tailored to adaptive planning and delivery.

While a limited number of didactic sessions provide foundational overviews of ART principles, most of the training is delivered through hands-on experience and one-on-one mentor-guided sessions. Fellows actively participate in ART clinical cases across multiple treatment platforms, where they are directly involved in executing adaptive workflows under the supervision of platform-specific mentors.

### 2.3. Clinical Resources

Online ART requires coordinated support from the clinical team, including attending physicians, therapists, dosimetrists, and physicists, as well as routine machine readiness processes and time-sensitive QA practices consistent with institutional standards. For the fellowship, an attending physicist serves as the primary rotation mentor and oversees training across multiple categories, including planning, online adaptive coverage, patient-specific QA, and workflow safety.

During online adaptive coverage, fellows are also paired with the covering physicist of the day. Training follows a staged model, beginning with observation and progressing to supervised hands-on participation based on competency milestones. The covering physicist documents progress using structured competency forms, which are reviewed in conjunction with the primary rotation mentor. Fellows also interact with the broader multidisciplinary team during adaptive sessions, and feedback from physicians and therapists is incorporated into longitudinal competency assessment.

Fellow scheduling is coordinated with clinical workload and determined by the primary rotation mentor to ensure that training exposure is meaningful without disrupting service coverage. Throughout training, attending-level oversight is maintained for clinical decision-making, supported by standardized checklists and defined supervision thresholds. Accordingly, the fellowship is designed to safeguard patient safety and minimize impact on throughput during peak hours. As part of program evaluation, we plan to prospectively log ART session durations and related workflow measures to quantify training-related effects on efficiency and safety over time.

### 2.4. Competencies and Evaluation Strategy

The MPART Fellowship incorporates a competency-based evaluation framework aligned with the six core competencies outlined by the Accreditation Council for Graduate Medical Education [16]: Patient Care, Medical Knowledge, Practice-Based Learning and Improvement, Systems-Based Practice, Professionalism, and Interpersonal and Communication Skills. Fellow performance is assessed through a structured, multi-dimensional process designed to align with clinical rotation milestones and programmatic learning objectives.

After each rotation block, fellows undergo tiered competency evaluations that assess their technical proficiency, comprehension of ART workflows, and clinical decision-making abilities. These evaluations are complemented by regular mentor check-ins, which provide opportunities for formative feedback, individualized coaching, and early identification of areas for improvement to support ongoing professional development.

### 2.5. Summarize of Planned Outcomes, Data Collection and Analysis Strategies

The overall planned outcomes, data collection, assessment and analysis strategies are outlined as the followings:Planned outcomes for future cohorts

For future cohorts, we plan to evaluate outcomes that capture both individual development and program impact. These will include individual-level competency progression and case volumes across rotations and modalities, as well as scholarly productivity such as abstracts, presentations, publications, and educational contributions. We will also track selected workflow and safety indicators, including ART session duration distributions and relevant incident-learning data, to better characterize how structured training interacts with real-world clinical performance.

2.Data collection methods

Data will be collected through standardized rotation logs and case logs completed by fellows and verified by mentors to ensure accuracy and consistency across platforms. We will also use structured mentor evaluation forms aligned with defined competency domains, supplemented by fellow surveys and annual program review reports. When feasible, key operational metrics—such as on-table time and other adaptive workflow markers—will be extracted from clinical systems to provide objective measures of efficiency and clinical integration.

3.Criteria for performance assessment

Performance assessment will be anchored to defined minimum exposure thresholds and competency levels expected by the end of each rotation and at graduation. We will also apply structured criteria for determining “independent coverage readiness” for ART-related activities, while maintaining attending oversight for clinical decision-making. This framework is intended to ensure that autonomy is earned through demonstrated competence rather than assumed based on time-in-rotation.

4.Planned analysis strategies

Our analysis will primarily rely on descriptive statistics to summarize case volumes, competency attainment, mentor evaluations, and workflow indicators across platforms and cohorts. Where feasible, we will incorporate within-fellow pre-/post-comparisons on selected measures to assess development over time. We also plan cautious descriptive comparisons to reference groups such as recent residency graduates who did not complete MPART, with explicit acknowledgment of the limited sample size and non-randomized nature of these early comparisons.

5.Longitudinal assessment of program effectiveness

Program effectiveness will be assessed longitudinally through annual aggregation and review of outcomes by the MPART Steering Committee, with iterative refinements to the curriculum, supervision model, and assessment tools as needed. In addition to near-term educational and workflow measures, we will track graduates’ subsequent roles—such as leadership in ART implementation or academic positions—as a longer-term indicator of the fellowship’s impact on the evolving adaptive radiotherapy workforce

## 3. Results

### 3.1. Overview of the MPART Program

The MPART Fellowship is a two-year advanced training program established at our center. It is designed to provide structured clinical and research experience for post-graduate radiation therapy physicists interested in developing expertise in personalized ART. The fellowship emphasizes mastery of cutting-edge ART technologies and clinical applications, with a focus on imaging-guided, platform-specific treatment strategies.

Program governance is provided by the MPART Steering Committee, a multidisciplinary body composed of the fellowship director, service leads for each ART modality, key clinical physicists actively engaged in adaptive therapy, physician representatives from high-ART-volume disease site teams, and liaisons from the medical physics residency and graduate programs. This committee is responsible for curriculum design, mentor assignment, rotation oversight, project evaluation, and overall quality assurance. Interaction with existing educational programs—including the CAMPEP-accredited residency, radiation oncology fellowships, and translational physics research initiatives—ensures synergy across training levels and fosters a collaborative, cross-disciplinary learning environment.

Fellows rotate through modular clinical blocks centered on three advanced delivery modalities available at our center: (1) CBCT-guided ART using Varian Ethos (Varian Medical System, Palo Alto, CA, USA), (2) MR-guided RT and ART via Elekta Unity (Elekta, Stockholm, Sweden), and (3) PET-guided radiotherapy using the RefleXion X1 platform (RefleXion Medical, Hayward, CA, USA). Figure 1 illustrates the advanced treatment modalities available within the department, which together offer a comprehensive exposure to CBCT-, MR-, and PET-guided RT and ART workflows.

For each rotation, fellows engage in a combination of structured didactics, supervised clinical learning, simulation/dry-run activities (when platform tools are available), and progressive hands-on case participation. The rotation mentor works with the fellow to develop a personalized training schedule aligned with rotation goals, baseline experience, and contemporaneous clinical workload. Didactic sessions are typically concentrated early in the rotation (e.g., one to two sessions per week during the first 2–3 weeks) to establish platform-specific principles, workflow expectations, and safety considerations, with additional sessions provided as needed based on performance and evolving clinical demands.

To broaden exposure beyond a single mentor and ensure consistency with real-world coverage models, fellows observe and shadow multiple covering physicists within the ART team across key tasks, including ART planning, online adaptive coverage, and patient-specific QA. For platforms that provide vendor emulators or equivalent training environments, fellows complete simulation cases or dry-run exercises early in the rotation and when new workflows are introduced, allowing practice of contouring, planning, and decision-making in a low-risk setting.

Clinical supervision follows a staged model. Using online adaptive coverage as an example, fellows begin with observation of multiple sessions, then perform discrete tasks under direct supervision of the covering physicist, and gradually transition to more advanced responsibilities as competency milestones are achieved. Competency is documented using platform-specific checklists and structured evaluation forms, with sign-off by the covering physicist and validation by the primary rotation mentor. Weekly mentor feedback and milestone reviews are incorporated within each rotation, and the program director meets with fellows regularly to support longitudinal progress.

Evaluation emphasizes both educational and clinical integration metrics, including the number and diversity of ART cases participated in across modalities, progression through defined competency levels, mentor ratings of technical skills, clinical judgment, communication, and professionalism, and feedback from multidisciplinary team members (physicians, therapists, dosimetrists) when relevant. We also clarified that fellows are expected to achieve defined minimum exposure and competency thresholds by the end of each rotation, with case experience spanning multiple disease sites and modalities aligned with institutional clinical volume and scheduling.

The program targets physicists who have either completed a CAMPEP-accredited residency or have at least two years of full-time clinical experience in radiation oncology but lack formal ART training. Both domestic and international applicants are encouraged to apply. The current cohort includes one fellow in active training, with a second fellow slated to begin later this year. Table 2 summarizes recruitment metrics to date with a total of 22 applicants—10 in 2023 and 12 in 2024. Of these, 36% hold an MS degree, and 64% have completed a PhD. The median number of years of prior clinical experience is 4.5 or more. Notably, international applicants constitute approximately 72% of the candidate pool, reflecting a strong global interest in ART and the limited availability of advanced adaptive platforms in many regions outside the United States. The selection criteria are based on the applicant’s motivation to develop ART skills, track record of clinical and research experience, communication skills, and commitment to the two-year training program.

### 3.2. Rotation Structure and Topics Covered

The MPART curriculum is delivered through modular clinical rotations structured around platform-specific workflows and guided by clearly defined learning objectives. Figure 2 presents a sample two-year rotation schedule. Each rotation blends didactic instruction with intensive, hands-on clinical training led by assigned rotation mentors. Fellows are immersed in day-to-day ART operations and are expected to actively participate in all phases of the adaptive workflow, including image review, target delineation, replanning, QA, and treatment delivery. The first year focuses on basic training and learning. In the second year, the fellows are expected to function at a near-attending level, although still under the supervision of a mentor physicist, and actively participate in all steps of clinical coverage. For example, Table 3 presents more specific educational goals related to the CBCT-guided ART rotation. Similar rotation objectives or topics are designed for each training module.

Training milestones are achieved primarily through daily clinical case involvement and regular mentor-mentee interactions. Routine evaluation check-ins and milestone reviews ensure fellows are progressing toward competency goals. Fellows are encouraged to engage in complex and novel cases to deepen their understanding and develop confidence in ART decision-making under real-world time constraints.

In addition to clinical rotations, each fellow is allocated protected time annually to pursue a development project aligned with ART research or operational needs. Recognizing the rapidly evolving nature of ART, the program encourages fellows to adopt a problem-solving mindset—identifying clinical gaps and proposing innovative solutions in collaboration with their mentors. Example project topics include AI-assisted auto-segmentation benchmarking, robust planning for ART under anatomical uncertainty, PET-guided multi-target optimization, automation of adaptive QA protocols, and workflow efficiency modeling. These projects reinforce core competencies while promoting translational impact and platform-specific innovation.

### 3.3. Evaluation and Competency Assessment

To promote reflective learning and reinforce clinical application, fellows are required to deliver a formal presentation at the end of the year. These presentations summarize rotation-specific objectives, practical challenges encountered, and contributions to clinical or research advancements. Table 4 presents an example of an evaluation table for competency, featuring multiple competency levels related to MR-guided treatment planning for ART. The competency is evaluated by rotation mentor based on feedback from the supervising physicist and other ART team members.

We designed evaluation metrics at two levels: educational/competency metrics (per fellow) and workflow/patient-care–related metrics (program level). These include case volume and diversity, progression through rotation-specific competencies, mentor evaluations, and scholarly output. We also incorporate safety- and efficiency-focused measures, such as ART-related incidents/near-misses captured through our institutional incident-learning system and prospective logging of adaptive session characteristics. The details are shown below:Educational/competency metrics (per fellow):
○Number and diversity of ART cases participated in and/or led per modality (e.g., planned case, covered adaptive session, chart check performed.)○Progression through competency levels as per rotation-specific checklists (e.g., similar to the MR-guided checklist in Table 4).○Mentor ratings on key domains (technical skills, decision-making, communication, professionalism).○Scholarly outputs (abstracts, presentations, publications, educational materials).
Workflow and patient-care–related metrics (at the program level):
○Number of ART cases covered by all fellows (planning, online session, chart checking).○Safety indicators (e.g., ART-related incidents or near-misses captured in the incident learning system, need for case abortion or offline re-review).○Feedback from other ART team members on overall fellow performance


Our rotation evaluation strategy is platform-specific, with separate competency checklists and case logs for CBCT-guided ART, MR-guided ART, and PET-guided RT/ART. We will analyze performance and exposure metrics per platform, acknowledging that positive transfer between platforms is a desired educational outcome rather than a pure confound. When reporting results, we will avoid attributing platform-specific effects unless supported by platform-stratified data.

The MPART Steering Committee, which includes both physicists and physicians, conducts semi-annual performance reviews, integrating mentor evaluations, fellow self-assessments, and progress toward programmatic goals. Additionally, fellows complete periodic satisfaction surveys to inform the continuous improvement of the training experience. If a fellow does not meet expected performance benchmarks, a structured remediation plan is initiated. This may include extended clinical rotations, supplemental mentorship, or targeted assignments designed to reinforce specific skill areas. The program remains committed to individualized development, ensuring that fellows are fully supported in achieving clinical and professional excellence in ART.

## 4. Discussion

The MPART Fellowship was established to address a critical gap in formal training for medical physicists in ART. Leveraging one of the most comprehensive and technologically advanced linac infrastructures in the country—including Varian Ethos (CBCT-guided ART), Elekta Unity (MR-guided ART), and RefleXion X1 (PET-guided BgRT)—the program offers immersive, platform-specific education to post-residency physicists. This fellowship is uniquely positioned to accelerate technical skill development, support multidisciplinary collaboration, and promote workflow harmonization across ART modalities.

The fellows are selected through a competitive process. The fellowship welcomes applicants from centers with and without access to online ART platforms, regardless of baseline ART experience, reflecting the program’s mission to provide structured, multi-platform training to a broad range of early-career physicists. We acknowledge the potential for selection bias toward highly motivated applicants and institutions with a strong interest in ART. Because the program remains in an early phase and only two fellows have been recruited to date, we do not attempt to make population-level inferences regarding representativeness from this initial cohort; rather, we present these data to characterize early interest and variability in baseline experience.

The program was intentionally structured to cultivate both clinical and academic leaders in ART. Fellows receive extensive training in treatment indications, planning techniques, delivery strategies, and quality assurance processes across multiple ART platforms. By the end of the two-year fellowship, participants are expected to become proficient not only in adaptive techniques but also in stereotactic body radiotherapy (SBRT), positioning them to advance precision radiation oncology practices nationally and internationally.

One of the most impactful aspects of the MPART Fellowship is its emphasis on direct, hands-on clinical involvement. Active participation in patient care enhances understanding of the complexities unique to ART, including the need for rapid image assessment, target adaptation, and time-sensitive decision-making. By engaging in direct execution of real clinical cases, adaptive planning, and clinical consultations, fellows bridge the gap between theoretical knowledge and practical application. This experiential learning model has proven essential for developing confidence and independence in managing adaptive workflows.

ART demands a high level of real-time collaboration between physicists, radiation oncologists, dosimetrists, therapists, and nurses. Unlike many traditional physics responsibilities—such as quality assurance or chart review, which are performed independently—ART places the physicist at the center of a dynamic, procedure-like process. Effective communication becomes critical, especially during plan review, adaptation, and treatment delivery. The fellowship helps fellows develop the interpersonal and communication skills necessary for this team-based environment, reinforcing their ability to lead and contribute in high-stakes, multidisciplinary clinical settings.

Some implementation challenges are worth mentioning here. Establishing and maintaining a specialized training program, such as MPART, requires significant institutional commitment. We are fortunate to have strong support from departmental leadership and administrative infrastructure, which has been critical for launching and sustaining the fellowship. For institutions with limited access to ART platforms or financial resources, partnering with academic centers for training collaborations or rotations may be a viable strategy. The successful training of an ART fellow depends in part on adequate case volume and access to multiple ART platforms. For smaller or resource-constrained centers, these factors may limit training breadth and effectiveness. In addition, differences in technology and vendor-specific workflows can introduce variability in experience across modalities. To mitigate this, we emphasize consistent training structures and shared competency goals across platforms, and we leverage mentors who cover multiple ART services to help standardize teaching and expectations. Mentorship also represents a significant investment. Rotation mentors play a pivotal role in the success of each fellow, and their clinical workload must be balanced with dedicated educational responsibilities. Encouraging fellows to contribute meaningfully to service lines—such as assisting with clinical cases—helps create a mutually beneficial model where training efforts also support departmental operations. Another notable challenge is the lack of standardized, comprehensive educational materials specific to ART. As ART remains a rapidly evolving field, few textbooks or curricula currently exist. However, the emergence of consensus guidelines, task group reports, and white papers focused on ART is beginning to address this gap, offering valuable resources for training programs. Additionally, workflows developed at a specific institution may not generalize easily to other settings. Fellows are encouraged to critically evaluate each workflow, identify institutional customizations, and consider alternative strategies that could be adapted in resource-constrained or differently structured clinical environments. Ultimately, the rapid pace of technological advancements in ART necessitates ongoing updates to the curriculum. Programs must remain agile and responsive to emerging practices, platforms, and evidence-based strategies. Finally, as this fellowship program is still in its early phase of establishment, the primary goal of this report is to introduce the structure, framework, and implementation model of the program. Comprehensive outcome analyses—such as quantitative assessments, fellow feedback, and longitudinal competency evaluations—will be incorporated in future work as additional cohorts complete training.

The MPART Fellowship was conceived as a standalone, structured educational pathway to address the growing clinical demand for ART-trained physicists. The program’s framework—including its governance model, rotation structure, mentorship philosophy, and project integration—may serve as a valuable template for other institutions seeking to develop similar training initiatives. The training content, duration, and objectives can be customized or tailored to align with institutional resources. Elements of this curriculum could also be adapted for incorporation into existing medical physics residency or graduate training programs. We want to clarify that MPART builds upon or fills the gap, rather than replaces CAMPEP residency training. Given the small numbers of fellows (1–2 per year), our primary objective is to demonstrate feasibility and describe educational outcomes, not to statistically prove superiority over CAMPEP residency or other training program. As such, formal power calculations for between-group comparisons are not feasible at this stage. In the future, we will descriptively compare selected outcomes (e.g., independent coverage readiness, ART case leadership, scholarly output) between MPART fellows and a reference group of recent residency graduates from our institution who did not complete MPART. Any such comparisons will be interpreted cautiously, with explicit acknowledgment of the limited sample size and non-randomized nature of the cohorts.

As the field of radiation oncology continues to shift toward personalized, adaptive treatment approaches, the need for dedicated ART training will only become more pressing. Programs like MPART offer a scalable model for developing the next generation of medical physicists who can lead this transformation with confidence and clinical excellence.

## 5. Conclusions

The MPART Fellowship addresses a critical gap in post-residency medical physics training by providing structured, hands-on experience in CBCT-, MR-, and PET-guided RT and ART. Through immersive clinical rotations, interdisciplinary collaboration, and translational research, the program prepares physicists to lead the implementation of personalized ART. Its framework may serve as a scalable model for other institutions aiming to advance precision radiation therapy through dedicated ART training. Future studies will focus on systematically collecting and analyzing feedback from graduated fellows, evaluating competency milestones, and correlating training outcomes with measurable professional or clinical impact.

## Figures and Tables

**Figure 1 healthcare-13-03315-f001:**
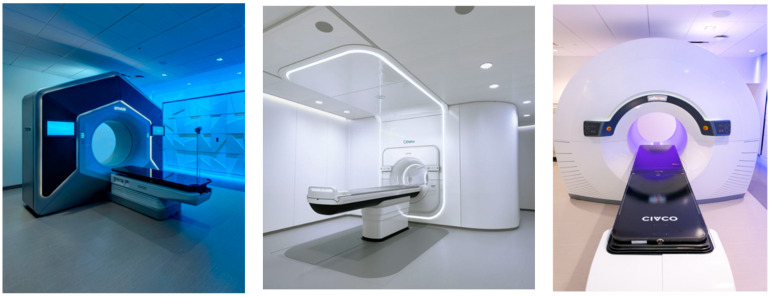
CBCT-guided (Varian Ethos), MR-guided (Elekta Unity) and PET-guided (RefleXion X1) treatment modality at our center. Currently Ethos and Unity support online ART.

**Figure 2 healthcare-13-03315-f002:**
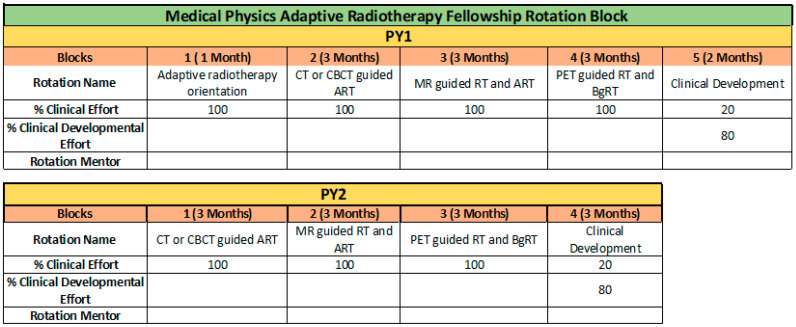
Sample rotation schedule for MPART fellow in the first and second year. Nine total rotation blocks.

**Table 1 healthcare-13-03315-t001:** General training objectives within the Medical Physics and Patient Care domain.

Medical Physics Knowledge Objectives	Clinical Knowledge Objectives
Learn pertinent scientific/clinical literature related to adaptive and personalized radiation therapy. Understand the physics and technology related to the realization of adaptive radiation therapy Master planning and quality assurance techniques on ART under different imaging guidance. Enhance skills of delivery and quality control of online ART. Strengthen the application of risk management and process improvement Pursue novel research opportunities in adaptive radiation oncology	Develop comprehensive ART physics skills across consultation, CT/MR/PET simulation support, personalized planning, safe delivery, machine and patient-specific QA, and multidisciplinary support. Strengthen physics decision-making for personalized ART, emphasizing imaging guidance, machine performance, plan robustness, and treatment monitoring. Build confidence in addressing complex physician requests and advising the team on imaging quality, adaptive/SBRT workflows, and potential complications.

**Table 2 healthcare-13-03315-t002:** Two-year recruitment statistics of applicants to MPART fellowship program.

Year	No. of Applicants	No. MS	No. PhD	Domestic	International	Median Year of Experience
2023	10	2	8	2	8	5+
2024	12	6	6	2	10	4+

**Table 3 healthcare-13-03315-t003:** Educational goals for the CBCT-guided ART rotation.

CBCT-Guided Adaptive Radiotherapy
Competency-Based Goals and Objectives This rotation covers CBCT-guided adaptive radiotherapy. Online adaptive radiotherapy is carried out on the Ethos system—an adaptive treatment planning system coupled with a ring gantry medical linear accelerator and high-quality CBCT. Overall Goals of this Rotation Understand the rationale, workflow, and key features of CBCT-guided adaptive radiotherapy in both offline and online settings. Understand the characteristics and learn to perform quality assurance tests on a CBCT-guided online adaptive platform—Ethos system Become proficient in designing robust treatment plans with CT or CBCT images for adaptive radiotherapy. Become proficient in performing patient-specific quality assurance on personalized adaptive treatment plans. Enhance skills in the successful delivery and quality control of an online adaptive radiotherapy treatment procedure. Develop a comprehensive understanding and skills regarding the risk management of adaptive radiotherapy treatment. Gain proficiency in responding to special consultation requests from physicians and other team members in tackling challenges related to imaging quality, treatment setup, axillary treatment device design, handling of implanted devices, treatment replanning, in vivo dosimetry, and any possible complications when executing CBCT-guided online or offline adaptation.

**Table 4 healthcare-13-03315-t004:** A sample competency checklist of MR-guided treatment planning for ART.

Level 1	Level 2	Level 3	Level 4	Level 5
Observe the treatment planning process for MR-guided radiotherapy and MR-guided adaptive radiotherapy. Recognize the key steps from initial MR and/or CT simulation to final plan generation. Understand major MR imaging sequences and their application in delineation, localization, and tumor tracking.	Practice key components in designing a robust adaptive treatment plan with MR guidance, including treatment simulation, image fusion, delineation, plan setup, plan optimization, dose calculation, and plan evaluation. Understand various treatment planning strategies and the selection of treatment techniques.	Develop robust, adaptive treatment plans for common treatment sites using appropriate treatment techniques with minimal assistance from faculty members.	Independently generate robust adaptive treatment plans for the most common treatment sites using appropriate treatment techniques.	Independently generate robust adaptive treatment plans or replans for all treatment sites, including challenging clinical cases requiring special considerations. Publish a research paper on a novel strategy of robust planning towards MR-guided radiotherapy or adaptive radiotherapy.

## Data Availability

The original contributions presented in this study are included in the article. Further inquiries can be directed to the corresponding author.

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
