# Peer review of "Medical Physics Adaptive Radiotherapy (MPART) Fellowship: Bridging the Training Gap in Online Adaptive Radiotherapy"

_healthcare, 2025, doi:10.3390/healthcare13243315_

Round 1

Reviewer 1 Report

Comments and Suggestions for Authors
  1. The introduction lacks information introducing the reader to AI in medicine. Please read and and cite the following article DOI: 10.3390/diagnostics13152582
  2. With 22 applications and 2 accepted candidates in 2023–2024, have you assessed the risk of selection bias and the representativeness of the cohort relative to centres without Ethos/Unity/RefleXion?
  3. What are the hard resource requirements for an ART session (acquisition time, staffing, daily/hourly QA, service windows) and how do they affect throughput and patient safety during peak hours?
  4. What measurable indicators of programme success will you plan (e.g., volume of ART sessions per fellow, median on-table time, percentage of replans accepted online, safety indicators) and with which reference group will you compare them to demonstrate superiority over standard CAMPEP training?
  5. How will you ensure adequate statistical power and minimise cross-platform learning effects?

Reviewer 2 Report

Comments and Suggestions for Authors

Thank you for the opportunity to review your manuscript describing the Medical Physics Adaptive Radiotherapy (MPART) Fellowship. The topic is timely and relevant, given the rapid expansion of online adaptive radiotherapy (ART) and the lack of formalized training pathways for medical physicists. The manuscript is clearly written, logically structured, and offers a valuable overview of a comprehensive educational initiative.

However, as a protocol article, the manuscript would benefit from additional methodological rigor and clarification. Below are major and minor comments intended to strengthen the clarity, reproducibility, and scientific value of the work.

Major Comments

Clarify the structure of the manuscript as a protocol - Although the paper is described as a protocol, it currently reads primarily as a narrative program description. To satisfy expectations for a protocol article, please add a dedicated and explicit section detailing: (i) the planned outcomes that future cohorts will be evaluated on, (ii) the data collection methods, including instruments, timing, and responsible parties, (iii) criteria for performance assessment, (iv) planned analysis strategies (qualitative or quantitative), (v) how program effectiveness will be measured over time.

This will make the protocol more transparent and replicable.

Include measurable training metrics - The manuscript describes learning goals but does not provide objective, quantifiable training milestones. Consider adding (i) expected case volumes per rotation, (ii) minimum competency requirements, (iii) examples of performance indicators (e.g., number of ART sessions led, QA tests performed, planning tasks completed), (iv) anticipated scholarly output.

Such information will clarify what “competency” means within the fellowship.

Better position the program within existing training frameworks - The introduction cites relevant AAPM/CAMPEP guidelines but does not discuss how MPART compares with other ART training initiatives (formal or informal). A brief review of comparable training models (short courses, vendor programs, institutional tracks) and how MPART differs or expands on them would strengthen the manuscript and contextualize its novelty.

Provide more detail on educational methodology - The manuscript specifies what is taught but not how. Please elaborate on (i) frequency and structure of didactic sessions, (ii) nature of real-time supervision during ART cases, (iii) use of simulation cases or dry-run exercises, (iv) evaluation rubrics used by mentors, (v) number/type of ART cases fellows are typically exposed to across modalities.

This detail is essential for reproducibility and for institutions wishing to emulate the program.

Expand the discussion of limitations and risks - You mention general implementation challenges, but the Discussion would benefit from a more explicit examination of: (i) dependence on high case volume and access to diverse ART platforms, (ii) potential variability in experience across modalities, (iii) rapid technological evolution and the need for curriculum updating, (iv) generalizability to resource-limited centers.

These additions will present a more balanced and realistic depiction of the program.

Minor Comments

-Edit for language clarity, conciseness, and typographical accuracy.

-A few sentences are repetitive or overly long; minor tightening would improve flow. Some typographical errors are present (e.g., "Medial Physics Education" should be "Medical Physics Education").

-Table 1 is dense; consider simplifying or converting to bullet points.

-Table 4 is useful but should include a short description of how competency levels are assessed and by whom.

-The figure 1provides limited scientific value. Consider integrating it with workflow diagrams or replacing it with something more informative (e.g., a schematic of the ART decision-making pathway).

-The statistics on applicants are informative, but interpretation is missing. You may comment on selection criteria, program capacity, expected career trajectories of fellows.

-Ensure all abbreviations are defined at first appearance.

Overall Recommendation

The manuscript presents an important and valuable training framework, but additional methodological detail and clarification are required for publication as a protocol.

Reviewer 3 Report

Comments and Suggestions for Authors

The manuscript "Medical Physics Adaptive Radiotherapy (MPART) Fellowship: Bridging the Training Gap in Online Adaptive Radiotherapy " is devoted to the description of a scholarship for teaching graduate students or practicing physicists advanced adaptive technologies and workflows of ART.

The manuscript should be rejected.

The structure and content of the manuscript do not correspond to the level of scientific publication and the level of the selected publication. The manuscript contains a description of the scholarship program for physics students, but there is no comparison with other existing programs with similar content.

No signs of exclusivity of the program are shown, except for providing access to the hardware. For example, studies of the effectiveness of the control group (without training) and the program participants are not presented. Or the comparability of the results of the participants of the master's level program or participants with an academic degree.

The manuscript looks more like an advertising description of a program than a scientific study.

Round 2

Reviewer 1 Report

Comments and Suggestions for Authors

The authors have complied with most of the reviewers' comments. They corrected the text of the article and thus contributed to its substantive value.

Reviewer 2 Report

Comments and Suggestions for Authors

I have carefully reviewed the revised manuscript. The authors have addressed most of my suggestions, and while there is still some room for improvement, the manuscript has now reached a level of maturity suitable for publication in a medium–impact factor journal. I fully support its publication without reservations.

Reviewer 3 Report

Comments and Suggestions for Authors

The manuscript can be accepted in its current form.